# Comprehensive Genome and Plasmidome Analysis of Antimicrobial Resistant Bacteria in Wastewater Treatment Plant Effluent of Tokyo

**DOI:** 10.3390/antibiotics11101283

**Published:** 2022-09-21

**Authors:** Tsuyoshi Sekizuka, Rina Tanaka, Masanori Hashino, Koji Yatsu, Makoto Kuroda

**Affiliations:** Pathogen Genomics Center, National Institute of Infectious Diseases, Shinjyuku-ku, Tokyo 162-8640, Japan

**Keywords:** urban sewage, effluent, *Enterobacterales*, ESBL, carbapenemase, Inc, plasmidome

## Abstract

To characterize environmental antimicrobial resistance (AMR) in urban areas, extended-spectrum β-lactamase- (ESBL)/carbapenemase-producing bacteria (EPB/CPB, respectively) from urban wastewater treatment plant effluents in Tokyo were isolated on CHROMagar ESBL plate. Complete genome sequence analysis, including plasmids, indicated that 126 CTX-M-positive isolates (31%) were identified among the 404 obtained isolates. The CTX-M-9 group was predominant (*n* = 65, 52%), followed by the CTX-M-1 group (*n* = 44, 35%). Comparative genome analysis revealed that CTX-M-27-positive *E. coli* O16:H5-ST131-*fimH41* exhibited a stable genome structure and clonal-global dissemination. Plasmidome network analysis revealed that 304 complete plasmid sequences among 85 isolates were grouped into 14 incompatibility (Inc) network communities (Co1 to Co14). Co10 consisted of primarily IncFIA/IncFIB plasmids harboring *bla*_CTX-M_ in *E. coli*, whereas Co12 consisted primarily of IncFIA(HI1)/Inc FIB(K) plasmids harboring *bla*_CTX-M_, *bla*_KPC_, and *bla*_GES_ in *Klebsiella* spp. Co11 was markedly located around Co10 and Co12. Co11 exhibited *bla*_CTX-M_, *bla*_KPC_, and *bla*_NDM_, and was mainly detected in *E. coli* and *Klebsiella* spp. from human and animal sources, suggesting a mutual role of Co11 in horizontal gene transfer between *E. coli* and *Klebsiella* spp. This comprehensive resistome analysis uncovers the mode of relational transfer among bacterial species, highlighting the potential source of AMR burden on public health in urban communities.

## 1. Introduction

The World Health Organization has endorsed a global action plan for antimicrobial resistance (AMR), which calls upon all nations to adopt mitigation strategies [1]. However, there is still a need to fully understand the ecology and evolution of AMR based on a one-health approach. In particular, the properties of the microbial resistome in ecosystems dominated by humans and how to monitor such environmental factors to evaluate their potential risk for promoting the evolution of AMR have to be sufficiently characterized. Instead of the limited nosocomial AMR surveillance, sewage AMR surveillance could highlight a broader picture of the global burden of AMR, including local-specific features, such as urban, suburban, industrial, and agricultural, based on the one-health approach.

There is a growing concern for sludge management due to the high levels of contaminants, and the design of current wastewater treatment plants (WWTPs) does not restrict the elimination of emerging contaminants and their metabolites [2]. These contaminants are released into rivers or streams as sewage effluents without interruption. The prevalence of AMR bacteria (ARB) and AMR genes (ARGs) in rivers would increase downstream of WWTPs because of effluents [3,4]. Indeed, the high density of bacteria in WWTPs could provide an optimum environment for horizontal gene transfer (HGT) between human pathogens and environmental bacteria [5]. Moreover, such environmental bacteria persist in the environment and later they might transfer back such genes to clinically relevant pathogens [6]. Therefore, wastewater and active sludge in WWTPs can act as reservoirs and environmental suppliers of antibiotic resistance, implying that they are hotspots for HGT under the selective pressure of antibiotics, disinfectants, and metals, even at low concentrations, enabling the dissemination of antibiotic resistance genes among different bacterial species. ARGs are located in multidrug-resistance (MDR) plasmids, which could have been transferred into broad recipient targets among different Proteobacteria strains, indicating a high possibility of HGT among bacteria in wastewater [7].

Thus far, the potential correlation between the integron *intI1* and anthropogenic pollution in WWTPs has been documented among the main hotspots of ARGs in the environment [8]. The importance of the synergy between plasmids and ARGs with flanking insertion sequences (ISs) has also been documented to potentiate the emergence of MDR-hypervirulent clones between WWTP- and human/animal-associated bacteria [5]. Ekanzala et al. reported that hospital wastewater showed significantly higher environmental resistome risk scores than all other assessed matrices [9,10], suggesting that hospital wastewater, effluent, and sewage sludge should be subjected to stringent mitigation measures to minimize such dissemination. Therefore, urban WWTPs are among the main sources of ARB and ARGs released into the environment [11]. However, the evolution of resistance and the spread of ARGs in WWTPs have not been widely researched and clearly evidenced.

Regarding a comprehensive global plasmidome analysis, Redondo-Salvo et al. characterized the global map of all publicly available over 10,000 reference prokaryotic plasmid sequences, including Proteobacteria, Firmicutes, Actinobacteria, Spirochaetes, Cyanobacteria, and Archaea [12,13]. Global analysis showed that more than 60% of the plasmids were grouped with host ranges beyond the species barrier, although plasmid transmission was constrained by taxonomic boundaries. Another report using long-read sequencing and an improved bioinformatics workflow for global sewage samples revealed marked AMR transmission among plasmids in complex untreated domestic sewage [13].

In this study, we characterized the complete genome sequences of extended-spectrum β-lactamase (ESBL)-or carbapenemase-producing bacteria (EPB or CPB) isolated from urban WWTP effluents in Tokyo. We have analyzed a marked genomic feature of CTX-M-positive *Enterobacterales* isolates that could be reflected in the urban areas of Tokyo as AMR surveillance. In addition, we determined the complete genome sequence of AMR plasmids in the EPB/CPB isolates to perform accurate gene network analysis (plasmidome analysis) to assess plasmid transmission among bacterial species.

## 2. Results

### 2.1. Bacterial Proportion in WWTP Effluents Genome Analysis of ESBL/Carbapenemase-Producing Bacteria (EPB and CPB) Isolated from WWTP Effluents and Environment

Sewage effluent samples were collected from eight WWTP sites along the Tama River, Tokyo Bay, and the recreational beach area (BEC1) [14]. EPB and CPB from fresh 50-mL samples of each effluent (Figure 1). CHROMagar ESBL selection demonstrated that potential clinically pathogenic *Enterobacterales* (*E. coli, Enterobacter, and Klebsiella* spp.) were observed in approximately 400 colonies from the effluents of WP4, WP5, and WP8 at all tested sampling times (Figure 1). More than a hundred typical chromogenic colonies, sufficient to characterize environmental pollution due to EPB and CPB, were isolated at each sampling time (Table 1). Whole-genome sequencing of all marked isolates (Appendix A) showed that 126 CTX-M-positive isolates were identified (31% in total; Table 1). The CTX-M-9 group (*bla*_CTX-M-9_, *bla*_CTX-M-13_, *bla*_CTX-M-14_, *bla*_CTX-M-24_, *bla*_CTX-M-27_, *bla*_CTX-M-65_, and *bla*_CTX-M-213_) was the predominant group with 68 isolates (54%), and the second was the CTX-M-1 group (*bla*_CTX-M-3_, *bla*_CTX-M-15_, and *bla*_CTX-M-55_) with 44 isolates (35%), following CTX-M-2 group (*bla*_CTX-M-2_) with 11 isolates (9%), and CTX-M-8 group (*bla*_CTX-M-8_) with three isolates (2%) (Appendix A). The EPB and CPB isolates on CHROMagar ESBL and its sequence type (ST) are summarized in Appendix A. Eighteen CPB isolates (six *bla*_IMP_-positive, eight *bla*_KPC_-positive, and two *bla*_NDM_-positive) were identified (Table 1).

### 2.2. E. coli ST131

Among the 71 *E. coli* isolates, ST131 was the predominant ST (*n* = 11 isolates) and CTX-M-27-positive *E. coli* ST131 (*n* = 9 isolates) was the most predominant, following CTX-M-3- or CTX-M-15-positive *E. coli* ST131 (*n* = 1 isolate each) (Appendix A).

Core-genome single-nucleotide variation (SNV) phylogeny (see the detailed procedures in Appendix A) of the ST131 isolates were compared with publicly available *E. coli* ST131 draft or complete genome sequences (315 strains; see Appendix A), suggesting that the four subclonal types of ST131 were primarily identified from the WWTP effluents, two isolates of ST131-*fimH41* with *bla*_CTX-M-27_, seven isolates of ST131-*fimH30* with *bla*_CTX-M-27_, one isolate of ST131-*fimH30* with *bla*_CTX-M-3_, and one isolate of ST131-*fimH30* with *bla*_CTX-M-15_ (Figure 2).

One isolate (WP5-S18-ESBL-09) of O16:H5-ST131-*fimH41* with *bla*_CTX-M-27_ exhibited marked clonality, showing less than 12 SNVs with clinical isolates from several countries (Southeast Asia, EU, and Oceania) (Figure 3A). Genome recombination and pan-genome analysis suggested that the genome structure of these strains was very stable, except for some ARGs that were transferred with conjugative plasmids (Figure 3B).

Three isolates of ST131-*fimH30* with *bla*_CTX-M-27_ (WP3-S18-ESBL-08, WP3-W18-ESBL-07, and WP3-W18-ESBL-09) exhibited marked clonality, with 8–13 SNVs to the clinical isolate SAMD00126441 in Japan (Figure 2), suggesting that a clinically potential subclone (O25:H4-ST131-*fimH30* with *bla*_CTX-M-27_) [18] has been detected in WWTP effluent in Japan.

### 2.3. Other E. coli ST Isolates

*E. coli* ST38 (*n* = 10 isolates), ST10 (*n* = 5 isolates), ST602 (*n* = 5 isolates), ST405 (*n* = 4 isolates), ST69 (*n* = 3 isolates), and ST648 (*n* = 3 isolates) were also detected in this study (Appendix A). The core genome SNV phylogeny for each ST isolate, including publicly available strains (Appendix A), suggests that CTX-M-14 is one of the major CTX-M variants in *E. coli* ST38, ST69, ST405, and ST648 (Figure 4).

For instance, all *E. coli* ST38 isolates (*n* = 10) were ST38-*fimH5*- and CTX-M-14-positive. However, these were classified into different serotypes (O86:H18 and O50/O2:H30), suggesting that at least two major serotypes of *E. coli* clones could play a pivotal role as hosts for *bla*_CTX-M-14_ gene dissemination in the global environment (Figure 4, panel ST38).

Regarding *E. coli* ST69 (*n* = 3 isolates), WP8-W19-ESBL-06 exhibited O15:H18-ST69-*fimH27* types and CTX-M-14 positivity, and its clonal dissemination in Europe and North America. In contrast, the other two isolates of ST69 (WP7-S17-ESBL-01 and WP8-S17-ESBL-12) exhibited CTX-M-14-positive but distinct serotypes (O25:H4 and O15:H18, respectively), and their relative host clones harbored different CTX-M genes, such as *bla*_CTX-M-15_ or *bla*_CTX-M-27_ (Figure 4, panel ST69). This indicated that these two CTX-M-14-positive ST69 isolates could be unique compared to other ST69 isolates in other countries, suggesting that the *bla*_CTX-M-14_ gene was recently acquired in *E. coli* ST69 in Japan.

With respect to *E. coli* ST405 (*n* = 4 isolates), most ST405 isolates, including publicly available genomes, exhibited O102:H6-ST405-*fimH27* and carried multiple CTX-M types. Two isolates (WP7-S18-ESBL-07 and WP7-S18-ESBL-09) exhibited distinct serotypes O45:H6 from other ST405 isolates. These two isolates showed the most similar genome SNV profile as SAMD00126438 (Human in Japan, Appendix A) and SAMD00076210 (sewage in Japan, Appendix A), suggesting that CTX-M-14-positive O45:H6-ST405-*fimH27 E. coli* isolate could be notable in Japan (Figure 4, panel ST405).

Regarding *E. coli* ST10 (*n* = 5 isolates), four isolates were CTX-M-positive, whereas all four carried distinct CTX-M genes (CTX-M-8, -14, -15, and -55). Most ST10 isolates, including those with publicly available genomes, exhibited O12:H4-ST10-*fimH27* and were isolated mainly from the Asian continent (Figure 4, panel ST10).

Regarding *E. coli* ST602 (*n* = 5 isolates), four isolates appeared to be clones because they were isolated from the same sample (WP2-W18-CRE-xx stands for isolate ID with meropenem selection at the WWTP WP2 site in winter 2018). The inclusion of publicly available ST602 genomes indicated that the CTX-M-27-positive O9:H9-ST602-*fimH86* isolate could be notable in Japan (Figure 4, panel ST602).

Regarding *E. coli* ST648 (*n* = 3 isolates), most ST648 isolates, including publicly available genomes, exhibited variable types, carrying mainly CTX-M-14 and -15 (Figure 4, panel ST648).

### 2.4. Other CTX-M Positive Bacteria

Among the 88 isolates of *Aeromonas* species (Table 1), three isolates of *A. hydrophila*, seven isolates of *A. caviae*, one isolate of *A. media*, and four isolates of *A. veronii* were identified as CTX-M producers (Appendix A). Although four isolates of CTX-M-14 producing *A. caviae* were detected in different WWTP effluents (WP2, 4, and 7), core-genome SNV phylogeny revealed that they were clonal in 21 SNVs with the same recombination and pangenome regions (Figure 5A), suggesting that the *A. caviae* clone may be successfully predominating in the general WWTP environment in Tokyo but not in WWTPs at specific locations.

Among the 48 isolates of *Klebsiella* species (Table 1), eight isolates of *K. pneumoniae*, seven isolates of *K. quasipneumoniae*, two isolates of *K. variicola*, and seven isolates of other *Klebsiella* spp. were identified as CTX-M producers (Appendix A). Core-genome SNV phylogeny revealed that four isolates of CTX-M-3-producing *K. quasipneumoniae* ST668 exhibited clonality in 14 SNVs and shared similar recombination and pan-genome regions (Figure 5B), although these isolates were obtained from the same place (WP5) but at different sampling times (summer 2017, summer 2018, and winter 2019), suggesting that the ST668 clone may remain in active sludge at WP5 for at least one year and more.

### 2.5. Plasmidome Analysis of EPB and CPB Isolated from WWTP Effluents and Environment

ESBL and carbapenemase β-lactamase genes were acquired by conjugative plasmid transfer among variable strains from the Proteobacteria group. Genetic network analysis among AMR plasmids could be useful to uncover the modes of relational transfer and trace dissemination. Of the 404 EPB/CPB isolates (Table 1), 85 were identified as complete genome sequences in this study (Appendix A). Moreover, out of the 304 complete plasmid sequences from the 85 strains described above, 73 β-lactamase-positive plasmids (Appendix A) were subjected to plasmidome network analysis based on orthologous analysis (see details in Appendix A). To characterize global ARG dissemination through plasmid transfer, 758 publicly available complete plasmid sequences showing a clear description of the isolation and source were selected from 19,904 complete plasmid sequences. A total of 831 complete plasmids (73 in this study and 758 in the public domain; see Appendix A) were subjected to pangenome analysis using Roary and NMDS (vegan in the R package), and the results revealed that 14 communities were classified and clustered (Figure 6).

Most network communities (Co) showed notable incompatibility (Inc) replicon-dependent distribution, suggesting that the network analysis was well-performed based on the genetic features of each Inc replicon. The major Co comprised Co10, Co11, and Co12 of the shared 423 plasmids among the tested 831 plasmids. Co10 was primarily IncFIA and IncFIB (AP001918) replicon plasmids harboring the *bla*_CTX-M_ gene in *E. coli* from human sources, whereas Co12 was primarily IncFIA(HI1) and Inc FIB(K) replicon plasmids harboring *bla*_CTX-M_, *bla*_KPC_, and *bla*_GES_ genes in *Klebsiella* spp. from human sources (Figure 6).

This network analysis highlighted the potential mutual role of Co11 plasmids between Co10 and Co12 plasmids because Co11 plasmids are placed around Co10 and next to Co12 (upper-left panel in Figure 6). Co11 is primarily rich in IncFII, Inc FII(pHN7A8), IncX1, and IncN replicon plasmids harboring *bla*_CTX-M_, *bla*_KPC_, and *bla*_NDM_ genes, and are mainly detected in *E. coli* and Klebsiella spp. from human and animal sources. Therefore, Co11 plasmids may contribute to the horizontal ARG transfer between *E. coli* and *Klebsiella* spp.

Regarding other notable co-exhibiting specific replicon and host bacteria, Co3 of comprised IncHI2 and IncHI2A replicon plasmids harboring *bla*_CTX-M_ and *bla*_IMP_ genes and was mainly detected in *Enterobacter* species (*E. cloacae*, 8; *E. hormaechei*, 6; *E. asburiae*, 3; others, 1) from human sources. Co4 comprised IncB/O/K/Z and IncI1-gamma replicon plasmids harboring the *bla*_CTX-M_ gene and were primarily detected in *E. coli* from human, environmental, and animal sources. Co6 comprised of an IncP6 replicon plasmid harboring *bla*_GES_ and *bla*_KPC_ genes and was primarily detected in *Aeromonas*, *Enterobacter*, and *Klebsiella* spp. from environmental sources.

The gene-specific distribution of carbapenemase genes was as follows: *bla*_IMP_ in Co1 and Co3; *bla*_KPC_ in Co6, Co8, and Co12 and *bla*_NDM_ in Co5, Co11, and Co13. Such ARG distribution may be involved in the plasmid replicon and its bacterial hosts.

## 3. Discussion

Here, we characterized WWTP-related EPB and its plasmids based on the complete genome sequences. Urban WWTP effluents could be a potential source of AMR burden. Therefore, there is concern that hospital [19] and community [20] effluents include a considerable proportion of ARGs in the environment. Among potential healthy EPB carriers, globally disseminated *E. coli* ST131 clones should first be investigated. This study suggests that multiple ST131 clones in WWTP effluents harbor the currently prevalent CTX-M variants (Figure 2). *E. coli* O16:H5-ST131-*fimH41* with *bla*_CTX-M-27_ was identified as a marked clone (Figure 3). Indeed, CTX-M-27-positive *E. coli* ST131 has been increasing in China [21], EU [22], Australia [23], New Zealand [23], and Japan [24], which has been speculated by the observation that CTX-M-27 might exhibit a higher hydrolyzing activity against ceftazidime compared with CTX-M-14 [25]. In contrast to ST131 with *bla*_CTX-M-27_, other *E. coli* STs still harbor *bla*_CTX-M-14_, which has been the predominant variant since the early 20th century in Japan [24], suggesting that the other STs have not shifted to acquire more potential CTX-M-27 thus far.

Plasmidome network analysis (Figure 6) exhibited a good approach for illuminating bacterial communication through plasmid-based HGT. This study highlights bacterial species-dependent plasmid tropism and compatibility among distinct species. The *bla*_CTX-M_ variants were identified in widely variable plasmids, whereas carbapenemase genes (*bla*_NDM_, *bla*_KPC_, and *bla*_IMP_) showed a typical plasmid replicon with a specific distribution, suggesting that the region-specific distribution of carbapenemase [26] may be associated with its locality [27]. In particular, specific *bla*_IMP_-containing integrons are markedly circulating among different bacteria in countries, such as Japan, Australia, and Thailand [27]. It has been reported that IMP-6-positive CPB clinical isolates are predominantly disseminated among chromosomally distinct isolates through the pKPI-6-related plasmid (IncN replicon [28]) in Japan [29]. Although IMP is the most predominant type of carbapenemase in clinical CPB in Japan, apart from clinical settings, we have reported the KPC-2 or NDM-5-positive WWTP-effluent isolates, i.e., the complete genome sequences of KPC-2-positive *Klebsiella pneumoniae* GSU10-3 [15], KPC-2-positive *Aeromonas hydrophila* GSH8-2 [16], KPC-2-positive *Aeromonas caviae* GSH8M-1 [16], and NDM-5- and CTX-M-55-coproducing *E. coli* GSH8M-2 [17] were determined. Overall, WWTP monitoring is an ideal approach to identify non-clinical but marked ARBs for the prediction of future clonal expansion from local to global dissemination.

According to the European Centre for Disease Prevention and Control, a significant proportion of antibiotics consumed by humans are in the community rather than in healthcare settings [30], suggesting that outpatient therapy could be the most effective factor in increasing the proportion of antimicrobials, selected ARGs, and ARB in WWTP. Therefore, a nation (or region)-wide survey of the resistome in WWTP effluents could provide a rapid and efficient method for assessing the environmental AMR burden in urban populations. To date, the healthy carriage rate of EPB has been rising worldwide. In Japan, the detection rate of EPB was reported to be 12.2% in healthy adult volunteers [31] and 15.6% in healthy food handlers [32]. As the number of EPB-positive healthy carriers has been increasing worldwide, Japan is no exception to the issue of increasing AMR healthy carriers. According to the Japan Nosocomial Infections Surveillance (JANIS), clinical reports of antibiotic-resistant gram-negative bacteria are increasing (https://janis.mhlw.go.jp/english/index.asp (accessed on 29 August 2022)). As suggested in this study, the whole genome sequence of the obtained ARBs (Table 1) and its plasmidome profiles in WWTP (Figure 6) could then reflect the structure and diversity of ARBs in the gastrointestinal tracts of urban residents within the WWTP catchment area.

The risk assessment of transmission of AMR vectors/reservoirs from the environment to humans cannot be adapted from the model for pathogens because most AMR vectors/reservoirs are supposed to be composed of non- or low-pathogenic bacterial species. As colonization may be asymptomatic in most humans, this may cause further dissemination under frequent abuse of antimicrobials in clinical use, leading to an underestimation of the extent of transmission of ARB from the environment to humans, as well as from humans to humans in the community. Therefore, they can colonize humans without notable symptoms, resulting in a healthy carrier [33]. Members of the family *Enterobacterales* and genera, such as *Aeromonas*, *Acinetobacter,* and *Pseudomonas* have been frequently documented as carriers of ESBL and carbapenemase genes in wastewater samples, and some of them may act as vectors for HGT [34]. Indeed, this study identified the clonal dissemination of CTX-M-14-positive *Aeromonas caviae* isolates at different WWTP sites (Figure 5A), indicating that such low-pathogenic *Aeromonas* spp. could be a potential reservoir for AMR dissemination. Therefore, reservoir ARBs should be considered in the overall AMR risk assessment.

## 4. Materials and Methods

### 4.1. Sample Collection

Sewage effluent samples were collected from eight WWTPs (WP1–WP9 and WP6 were treated as missing numbers owing to incorrect sampling locations) along the Tama River and around Tokyo Bay (Appendix A). Surface water from a recreational beach (BEC1) was used as the environmental control sample (Appendix A). Sampling was conducted during the summer and winter seasons for 2 years between August 2017 and February 2019 (Appendix A). The sampling procedure is summarized in Figure 1. A fresh 50 mL of effluent was centrifuged at 7000× *g* for 3 min, and the cell pellet was resuspended with residual water, followed by spreading all suspensions on CHROMagar ESBL plates (CHROMagar, Paris, France). The plates were then incubated at 36 °C for 24 h (Figure 1). Suspected colonies of *Escherichia coli*, *Klebsiella*, *Enterobacter*, *Aeromonas,* and *Pseudomonas* (Figure 1) were isolated as single clones, followed by whole-genome sequencing. In addition to EPB, CPB was isolated as described previously [15]. Briefly, 500 mL of effluent was filtered, and the membrane was incubated in 20 mL of Luria–Bertani (LB) broth supplemented with 1 mg/L meropenem at 37 °C for 14 h, after which the culture was spread on CHROMagar ESBL plates (CHROMagar, Paris, France).

### 4.2. Whole-Genome Sequencing of Bacterial Isolates

To determine the complete genome sequence, long-read sequencing data of isolated bacterial genomes were produced with more than 100-fold coverage and assembled using Flye version 2.5 [35]. Primary error correction using long reads was performed using Minimap version 0.2-r124 [36] and Racon version 1.1.0 [37], followed by circularization using Circlator version 1.5.3 [38]. The remaining error of the tentative complete circular sequences was corrected using Pilon version 1.18 with Illumina short reads [39]. Short plasmids (<10 kb) were assembled using A5-miseq version 20140604, using unmapped short reads against chromosomal and long plasmid sequences. The draft genome sequence was assembled using A5-miseq with only Illumina short-read data. Gene annotation was performed using DFAST version 1.2.3 [40] with the following databases: DFAST default database, ResFinder database [41], bacterial antimicrobial resistance reference gene (BARRG) database (PRJNA313047), and Virulence Factors Database [42]. Multilocus sequence typing (MLST) was performed using “mlst” program version 2.16.2 (Seemann T, mlst Github https://github.com/tseemann/mlst (accessed on 29 August 2022)) with PubMLST database (https://pubmlst.org/ (accessed on 29 August 2022)). Plasmid replicon typing was performed using the ABRicate program version 0.3 (Seemann T, Abricate, Github https://github.com/tseemann/abricate (accessed on 29 August 2022)) with the PlasmidFinder database [43].

### 4.3. SNV Phylogenetic Analysis and Pan-Genome Analysis

A flowchart of the single nucleotide variation (SNV) phylogenetic analysis is summarized in Appendix A. Assembly data and/or Illumina raw reads of four species (*E. coli*, *Aeromonas caviae*, and *Klebsiella quasipneumoniae*) were retrieved from the NCBI database. If assembly data were not deposited in the NCBI assembly database, the raw sequence data downloaded from the SRA database were assembled using SKESA version 2.3.0 [44]. MinHash sketch for each genome sequence in this study (*n* = 53) and NCBI data (*n* = 76,449) were constructed using the sourmash program version 2.0.0a4 [45] with the following parameters: compute -k 21 -scaled 4000. The MinHash sketch database was built using the sourmash program (parameter: index -k 21) with the NCBI dataset, followed by a MinHash search against 53 genome sequences. The collected dataset was analyzed for core genome SNV and pan-genome analyses for each species or sequence type (ST). In SNV analysis, the longest chromosomal sequences from this study were selected as the reference sequences. Repeat and prophage regions of the reference sequences were analyzed using NUCmer (MUMmer 3.0) [46] and prophet [47] programs, respectively. If the available data were only contig sequences from the NCBI database, the SimSeq program (https://github.com/jstjohn/SimSeq (accessed on 1 April 2018)) was used to construct simulated 150-mer paired-end reads with a 200 bp insert length. Read mapping was performed using BWA-MEM [48] with default parameters against reference chromosomal sequences, followed by variant calling using VarScan version 2.3.4 [49]. SNVs on repeat and predicted prophage regions were removed, and recombination regions were predicted using the Gubbins software [50], followed by filtering SNVs on recombination regions. Core genome SNV phylogenetic analysis was performed by the approximate maximum likelihood phylogenetic method using FastTree v2.1.10 [51], followed by visualization using Fandango version 1.3.0 [52] and interactive tree of life (iTOL) version 3 [53]. SNV network analysis was performed using the median joining network method and TCS network method of PopART (http://popart.otago.ac.nz (accessed on 29 August 2022)). Pan-genome analysis of predicted open reading frames (ORFs) was performed using Roary version 3.12.0 with default parameters [54].

### 4.4. Plasmidome Analysis

The flowchart for the plasmidome phylogenetic analysis is summarized in Appendix A. Complete plasmid sequences were retrieved from the NCBI database as of November 2019, followed by re-annotation and ARG detection using DFAST with the ResFinder and BARRG databases. A plasmid database of the nucleotide and protein sequences was constructed using BLAST. For plasmidome analysis, complete plasmids in the public database were selected according to the following criteria: (i) ≥90% nucleotide sequence identity and ≤1 × e^−100^ e-value against β-lactamase gene-positive complete plasmids revealed in this study, (ii) sharing ≥10 ORFs (≥99% identity) against these plasmids, (iii) presence of ARGs, and (iv) existence of metadata related to the isolation source. Pan-genome analysis using the collected plasmids was performed using Roary version 3.12.0 [54] with a parameter of 99% BLASTP percentage identity cutoff, followed by the construction of a distance matrix using the R package “proxy” with edge weights of plasmids sharing ≥ 10 ORFs. NMDS and community structure analysis were performed by “vegan” and “igraph” of R package, respectively. Plasmid data were visualized using Cytoscape version 3.7.2.

## 5. Conclusions

Urban WWTP effluents, even with proper treatment, may cause AMR burden with a high frequency of acquired ARGs in the environment. The dissemination of ARB/Gs in the environment might increase the risk of infectious diseases [55], but there is little direct evidence regarding their epidemiological effects. This study revealed resistome analysis based on complete genome sequencing and subsequent plasmidome analysis of EPB/CPB isolated from WWTP effluents, suggesting that every urban community, including hospitals, healthy carriers, and travelers, can be a potential source of ARGs. WWTP is speculated to be hotspots where various bacteria can acquire ARGs via HGT. Therefore, resistome analysis of AMR plasmids and their specific bacterial hosts in WWTP effluent is expected to identify the presence of undetected nosocomial infections, leading to the detection of potential ongoing dissemination in the overall community.

## Figures and Tables

**Figure 1 antibiotics-11-01283-f001:**
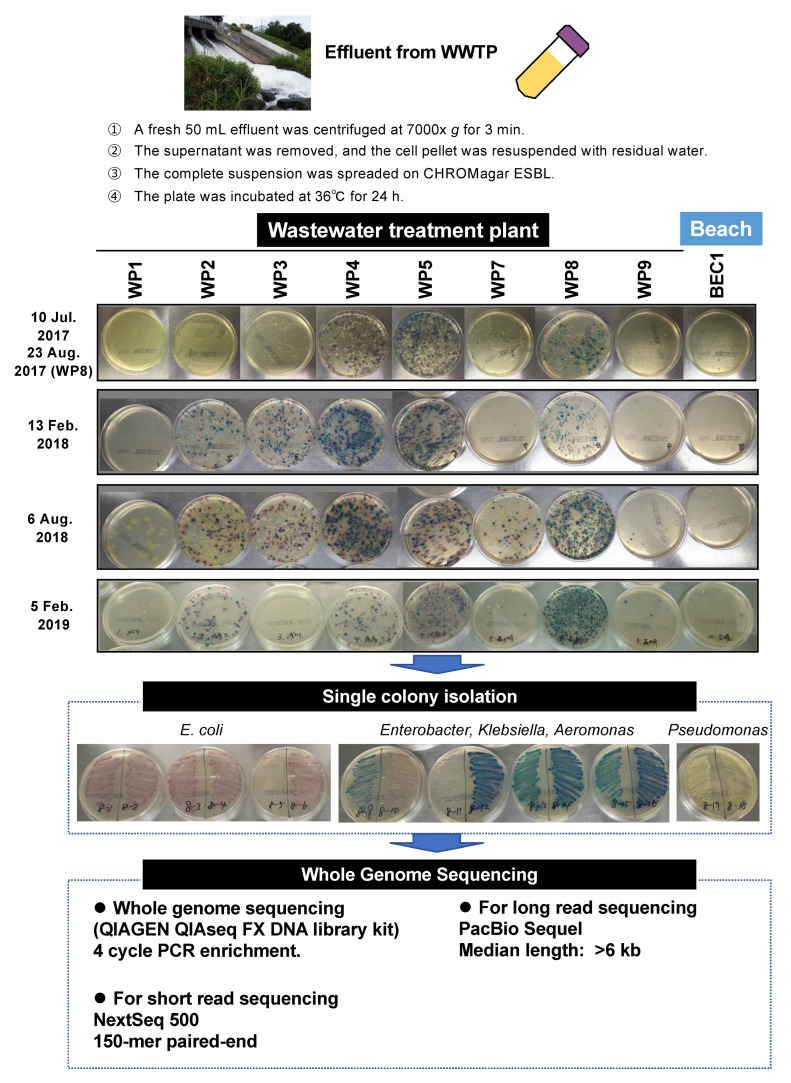
Experimental procedures to isolate potential ESBL-producing bacteria. Sewage effluent samples were collected from eight WWTPs (WP1–WP9, and WP6 was treated as missing numbers owing to incorrect sampling locations) along the Tama River and around the Tokyo Bay (Appendix A). Surface water from a recreational beach (BEC1) was used as an environmental control sample. Sampling was conducted during the summer and winter seasons for 2 years between August 2017 and February 2019 (Appendix A).

**Figure 2 antibiotics-11-01283-f002:**
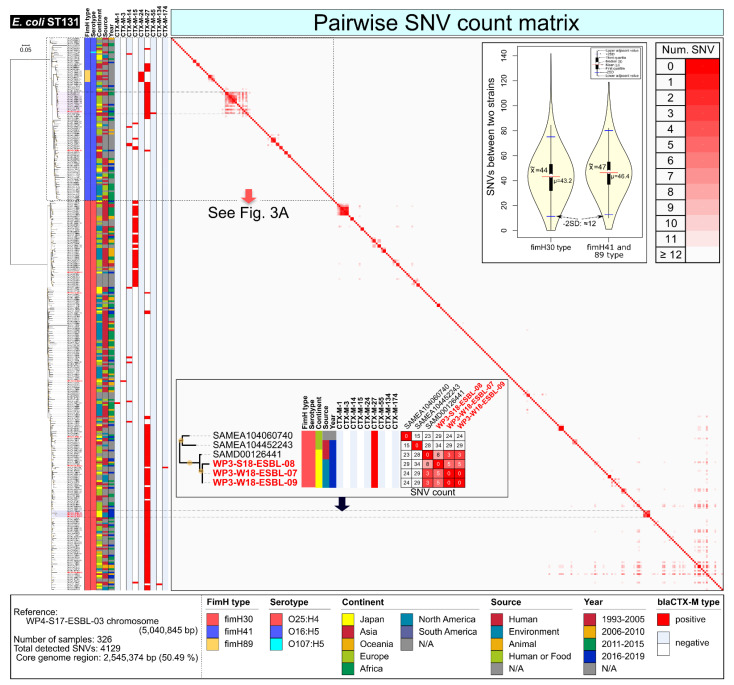
Core-genome SNVs phylogeny of *E. coli* ST131 isolates using publicly available ST131 draft or complete genome sequences (total 315 strains, see Appendix A). Isolates in this study was highlighted by red in the strain name (see detail in Appendix A).

**Figure 3 antibiotics-11-01283-f003:**
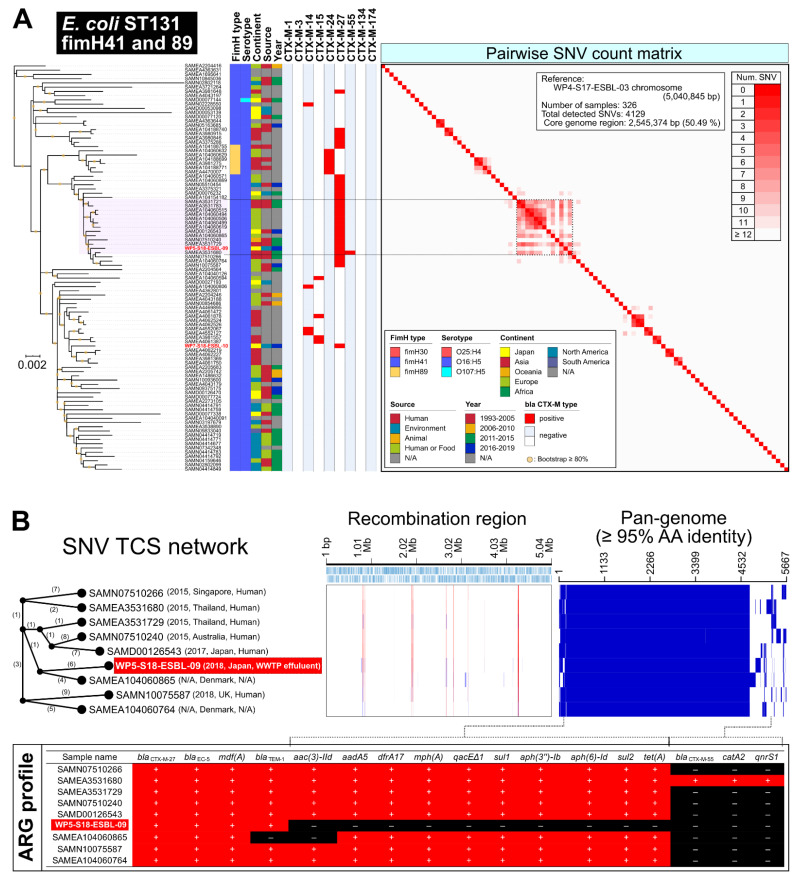
Characterization of *E. coli* O16:H5-ST131 isolates from WWTP effluents. (**A**) Highlight image for *E. coli* O16:H5-ST131-*fimH41* with *bla*_CTX-M-27_ from Figure 2. (**B**) Core genome SNV network, genome recombination, and pan-genome analysis of clonal O16:H5-ST131-*fimH41* with *bla*_CTX-M-27_ strains. Isolates in this study were highlighted by red in the strain name (see detail in Appendix A).

**Figure 4 antibiotics-11-01283-f004:**
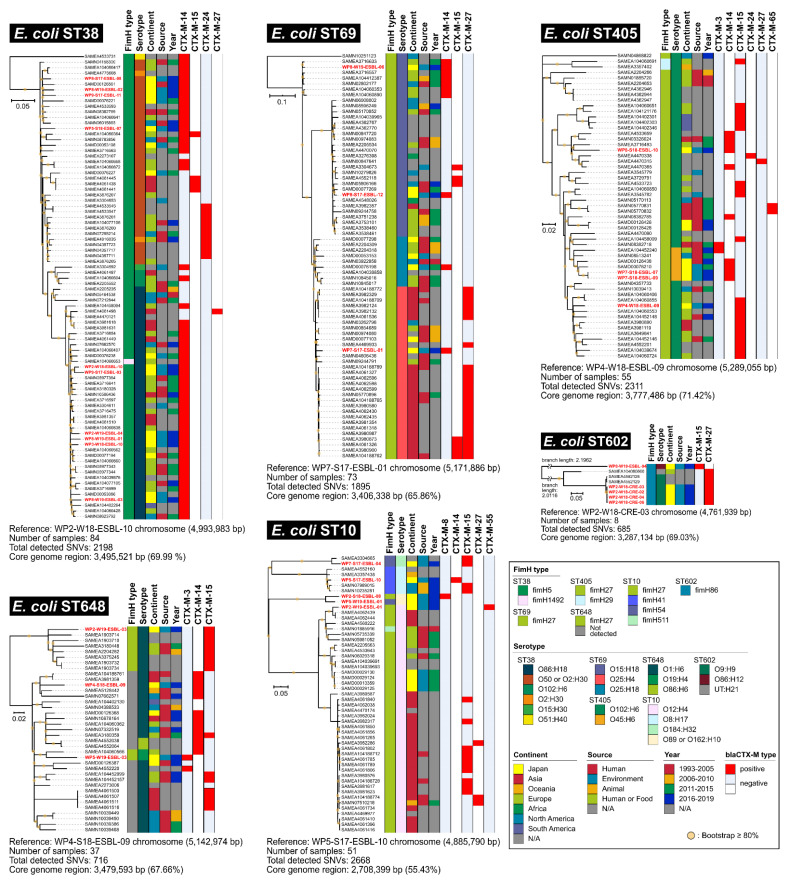
Core genome SNV phylogeny of other *E. coli* ST isolates from WWTP effluents. Isolates in this study was highlighted by red in the strain name (see detail in Appendix A).

**Figure 5 antibiotics-11-01283-f005:**
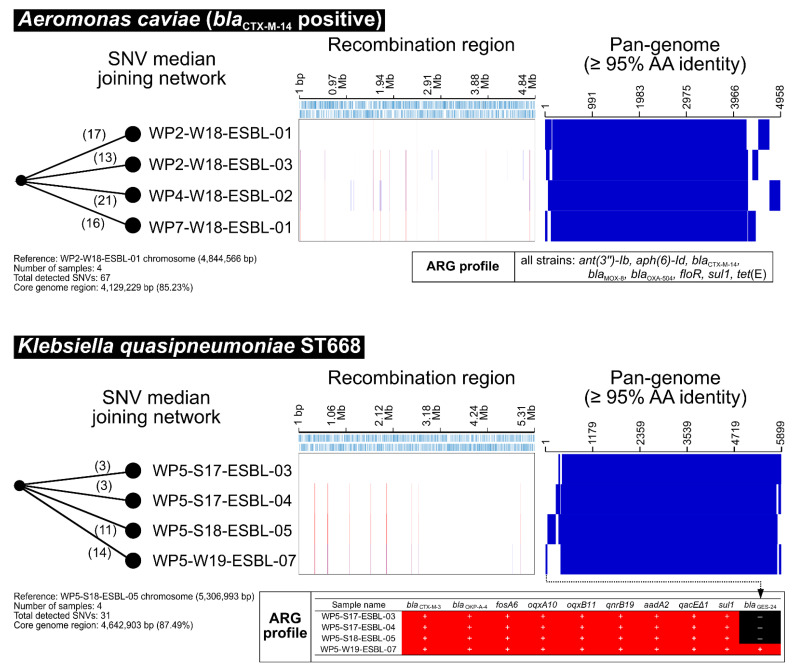
Characterization of other CTX-M-positive bacteria from WWTP effluents. Core genome SNV network, genome recombination, and pan-genome analysis of (**A**) *A. caviae* and (**B**) *K. quasipneumoniae* isolates.

**Figure 6 antibiotics-11-01283-f006:**
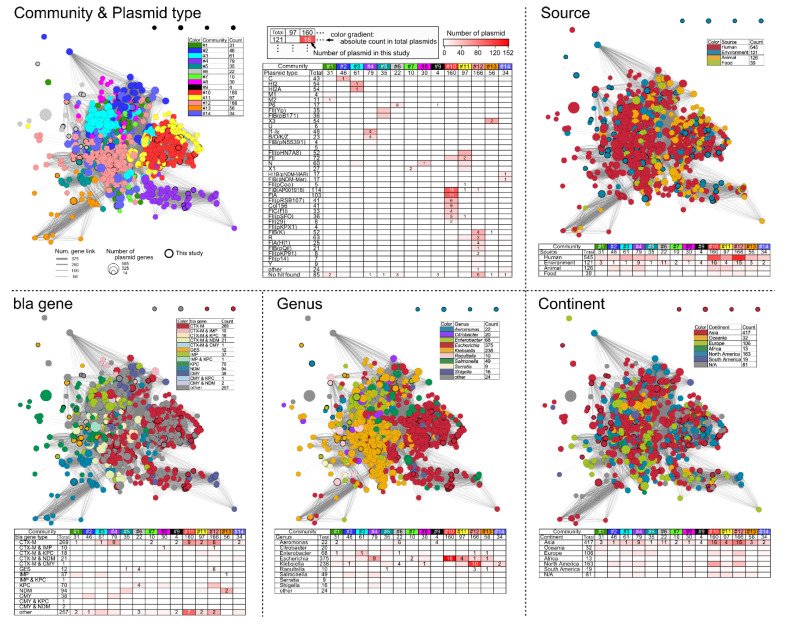
Plasmidome analysis of EPB and CPB isolates from WWTP effluents and environment. Nodes on the network are illustrated with different colors based on communities with sequence similarity, source, *bla* gene type, host bacteria genus, and its isolate location on the continent. The absolute counts for plasmids in plasmidome analysis are summarized in Appendix A.

**Table 1 antibiotics-11-01283-t001:** Whole-Genome Sequencing of CHROMagar ESBL-Positive Isolates Obtained from WWTPs.

		Summer, 2017	Winter, 2018	Summer, 2018	Winter, 2019	Total
Total isolates		98	93	109	104	404
EPB/CPB						
	*E. coli*	18	19	18	16	71
	*Klebsiella* spp.	8	12	18	10	48
	*Enterobacter* spp.	5	9	5	4	23
	*Acinetobacter* spp.	10	4	11	1	26
	*Pseudomonas* spp.	24	12	20	25	81
	*Aeromonas* spp.	17	25	25	21	88
	Others	16	12	12	27	67
β-lactamase type						
	CTX-M	29	35	32	30	126
	IMP	0	2	2	2	6
	KPC-2	0	0	5 ^a^	3	8
	NDM	0	0	1 ^b^	1	2
	GES	1	5	4	6	16

^a^ [15,16]; ^b^ [17]; See the information of whole-genome sequence for all isolates in Appendix A. EPB, extended-spectrum β-lactamase-producing bacteria CPB, carbapenemase-producing bacteria.

## Data Availability

All complete sequences are available in the DDBJ/EMBL/GenBank database (accession numbers AP021908–AP022304; see Appendix A). All raw read sequence files are available from the DRA/SRA database (accession numbers DRR199157–DRR199560 [whole-genome data, see Appendix A).

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
