# Peer review of "Comprehensive Genome and Plasmidome Analysis of Antimicrobial Resistant Bacteria in Wastewater Treatment Plant Effluent of Tokyo"

_antibiotics, 2022, doi:10.3390/antibiotics11101283_

Round 1

Reviewer 1 Report

This article titled “comprehensive genome and plasmidome analysis of AMR bacteria in wastewater treatment plant effluent of Tokyo”, describes resistome analysis based on complete genome sequencing and subsequent plasmidome analysis of EPO/CPO isolated from WWTP effluents in Tokyo, Japan. This work showed that every urban community (healthy carriers, hospitals and travelers) can act as a potential source of antibiotic-resistance genes. Authors should take into account some points as follows:  

1. Title: please replace AMR by antimicrobial resistant

2. Please replace organism by bacteria in all of the manuscript.

3. Please add spp. for Klebsiella

4. Bacterial species should be in italic.

5. Introduction: authors may add references in some points. Some parts are not referenced. So, please, review it.

6. Line 89-93: add the remaining CTX-M variant identified (from 20 isolates).

7. Please insert the figure 1 in materials and methods section.

8. Figure 1: The authors should mention that BEC1 was used as an environmental control sample.

9. In all figures, there is no need to explain the methodology or the result. Please, remove these sentences from the caption.

10. Line 115: It will be more useful to state on the other detected CTX-M variant in E. coli ST131.

11. Line 262: please precise each Enterobacter species.

12. Line 291-295: Please rephrase.

13. Change Enterobacteriaceae by Enterobacterales.

14: Line 342: Please rephrase (some of them may act as vectors).

15. Line 342: have been frequently documented as carriers in wastewater samples is inaccurate and should be replaced with have been frequently documented as carriers of ESBL and carbapenemase genesor another appropriate wording.

16. Line 358: which were the criteria used for selecting these species: E. coli, Klebsiella, Enterobacter, Aeromonas and Pseudomonas since that the aquatic environment carried a large variety of bacterial species.

 17. Line 360: Could the authors explain why they used the filtration method for the detection of ESBL and carbapenemase-producing bacteria from wastewater samples known by their high concentration and the presence of suspended solids. 

Author Response

Comments and Suggestions for Authors

This article titled “comprehensive genome and plasmidome analysis of AMR bacteria in wastewater treatment plant effluent of Tokyo”, describes resistome analysis based on complete genome sequencing and subsequent plasmidome analysis of EPO/CPO isolated from WWTP effluents in Tokyo, Japan. This work showed that every urban community (healthy carriers, hospitals and travelers) can act as a potential source of antibiotic-resistance genes. Authors should take into account some points as follows: 

  1. Title: please replace AMR by “antimicrobial resistant”

Response:

As suggested by reviewer, it was revised to ‘antimicrobial resistant’.

  1. Please replace “organism” by “bacteria” in all of the manuscript.

Response:

As suggested by reviewer, it was revised to ‘bacteria’. As following, EPO/CPO was revised to ‘EPB/CPB’.

  1. Please add spp. for Klebsiella

Response:

As suggested by reviewer, it was revised to ‘Klebsiella spp.’.

  1. Bacterial species should be in italic.

Response:

All bacterial species was revised in italic.

  1. Introduction: authors may add references in some points. Some parts are not referenced. So, please, review it.

Response:

Two references were additionally cited in the Introduction. We appreciate the suggestion.

Fijalkowski, K.; Rorat, A.; Grobelak, A.; Kacprzak, M.J. The presence of contaminations in sewage sludge - The current situation. J Environ Manage 2017, 203, 1126-1136, doi:10.1016/j.jenvman.2017.05.068.

Bengtsson-Palme, J.; Kristiansson, E.; Larsson, D.G.J. Environmental factors influencing the development and spread of antibiotic resistance. FEMS Microbiol Rev 2018, 42, doi:10.1093/femsre/fux053.

  1. Line 89-93: add the remaining CTX-M variant identified (from 20 isolates).

Response:

Remaining CTX-M variants were included in the text as follows:

‘The CTX-M-9 group (blaCTX-M-9, blaCTX-M-13, blaCTX-M-14, blaCTX-M-24, blaCTX-M-27, blaCTX-M-65, and blaCTX-M-213) was the predominant group with 68 isolates (54%), and the second was the CTX-M-1 group (blaCTX-M-3, blaCTX-M-15, and blaCTX-M-55) with 44 isolates (35%), following CTX-M-2 group (blaCTX-M-2) with 11 isolates (9%), and CTX-M-8 group (blaCTX-M-8) with 3 isolates (2%) (Table S2).’

  1. Please insert the figure 1 in materials and methods section.

Response:

Figure 1 includes the image of actual ESBL-producing isolates on CHROMagar ESBL plate. Therefore, it can be stayed in the result section at first presentation.   

  1. Figure 1: The authors should mention that BEC1 was used as an environmental control sample.

Response:

The legend in Figure 1 includes the description of ‘BEC1 was used as an environmental control sample’. What information should be included further to meet the request?

  1. In all figures, there is no need to explain the methodology or the result. Please, remove these sentences from the caption.

Response:

As suggested by reviewer, redundant sentences with the results section were removed from the caption in each Figure.

  1. Line 115: It will be more useful to state on the other detected CTX-M variant in E. coli ST131.

Response:

Remaining 2 isolates were included in the text as below,

‘and following CTX-M-3- or CTX-M-15-positive E. coli ST131 (n = 1 isolate each)’.

  1. Line 262: please precise each Enterobacter species.

Response:

Enterobacter species was included for community Co3 as below,

‘Co3 of comprised IncHI2 and IncHI2A replicon plasmids harboring blaCTX-M and blaIMP genes and was mainly detected in Enterobacter species (E. cloacae, 8; E. hormaechei, 6; E. asburiae, 3; others, 1) from human sources.’

  1. Line 291-295: Please rephrase.

Response:

The sentence was rephrased to

‘Indeed, CTX-M-27-positive E. coli ST131 has been increasing in China [16], EU [17], Australia [18], New Zealand [18], and Japan [19], which has been speculated by the observation that CTX-M-27 might exhibit a higher hydrolyzing activity against ceftazidime compared with CTX-M-14 [20].’.

  1. Change Enterobacteriaceae by Enterobacterales.

Response:

As suggested by reviewer, it was revised to ‘Enterobacterales’.

14: Line 342: Please rephrase (some of them may act as vectors).

Response:

As suggested by reviewer, it was revised.

  1. Line 342: “have been frequently documented as carriers in wastewater samples” is inaccurate and should be replaced with “have been frequently documented as carriers of ESBL and carbapenemase genes” or another appropriate wording.

Response:

As suggested by reviewer, it was revised to add ‘of ESBL and carbapenemase genes’.

  1. Line 358: which were the criteria used for selecting these species: E. coli, Klebsiella, Enterobacter, Aeromonas and Pseudomonas since that the aquatic environment carried a large variety of bacterial species.

Response:

There is no intension to isolate a specific bacteria species, but pink and dark-blue colonies for E. coli and Klebsiella were first priority to pick up for the isolation. Indeed, many Aeromonas, Acinetobacter, Stenotrophomonas species were isolated in this study.

  1. Line 360: Could the authors explain why they used the filtration method for the detection of ESBL and carbapenemase-producing bacteria from wastewater samples known by their high concentration and the presence of suspended solids.

Response:

At first, we used WWTP effluents, not but influents.

Most people are rarely exposed by WWTP influent, whereas urban WWTP effluents, even with proper treatment, may cause AMR burden with a high frequency of acquired ARGs in the environment. Thus, we considered that the effluent is one of the best environmental sources to investigate the actual risk assessment for people in river and ocean nearby WWTP.

Basically, WWTP effluent was treated by A2O (anaerobic-anoxic-oxic) and chlorination, filtration was needed to obtain carbapenemase-producing bacteria. 

Reviewer 2 Report

Sekizuka et al analyzed comprehensively the genome and plasmidome of AMR Bacteria in Tokyo WWTP effluent samples. It has done excellent experiments but the presentation was average and could be done better way than now. Majorly the paper has a more technical side than public health implications. Slightly they could balance public health implications for their finding too.

I have a few proposals for consideration. If you think, my proposal can improve the quality of your paper, you can accept them.

Line 10- -or can be replaced with and, as ESBL and CPO are slightly different.

Line 11- It can be great to provide one sentence about your methodology, what was selective media etc

Line 11- Determination ? or analysis

Line 12- suggested better to say indicated or found?

Line 13- is it among CTX-M ?

line 24- it can be great to have one concluding sentence with public health implications

line 44- I think it demands something like this: Such environmental bacteria persist long in the environment and later they can transfer back such genes to clinically relevant pathogens

Line 67: can you add from this paper also- Citation Kirstahler P, Teudt F, Otani S, Aarestrup FM, Pamp SJ. 2021. A peek into the plasmidome of global sewage. mSystems 6: e00283-21. https://doi.org/10.1128/mSystems .00283-21.

Line 160- is it the most predominant subclonal type?

Table 1- what are these a,  b citations please can you open them? Do not use abbreviations in the table open EPO and CPO in the footnote.

Figures 1-4: seems like a little crowded, if any of your figures are not relevant in the main text you can push them in supplemental (it is just optional).  If some of them are based on secondary reads from databases.

Line 160- delete also

He concluded WWTP effluent is expected to identify the presence of undetected nosocomial infection, ongoing in the community. But, if we want to see from the perspective of wastewater-based surveillance, it supposes to use wastewater influent. Please mention that somewhere in the discussion or the introduction (here this study can be a potential reference DOI: 10.3389/fmicb.2022.887888.

Author Response

Comments and Suggestions for Authors

Sekizuka et al analyzed comprehensively the genome and plasmidome of AMR Bacteria in Tokyo WWTP effluent samples. It has done excellent experiments but the presentation was average and could be done better way than now. Majorly the paper has a more technical side than public health implications. Slightly they could balance public health implications for their finding too.

I have a few proposals for consideration. If you think, my proposal can improve the quality of your paper, you can accept them.

Line 10- -or can be replaced with and, as ESBL and CPO are slightly different.

Response:

It was revised to ‘extended-spectrum β-lactamase- (ESBL)/carbapenemase-producing bacteria (EPB/CPB, respec-tively)’

Line 11- It can be great to provide one sentence about your methodology, what was selective media etc

Response:

The phrase ‘on CHROMagar ESBL plate.’ was added in the abstract.

Line 11- Determination ? or analysis

Response:

It was revised to ‘Complete genome sequence analysis’.

Line 12- suggested better to say indicated or found?

Response:

It was revised to ‘indicated’.

Line 13- is it among CTX-M ?

Response:

It means that ‘among the 404 obtained isolates’.

line 24- it can be great to have one concluding sentence with public health implications

Response:

Last sentence in the abstract was revised, as reviewer suggested.

line 44- I think it demands something like this: Such environmental bacteria persist long in the environment and later they can transfer back such genes to clinically relevant pathogens

Response:

The point was added in the introduction, as reviewer suggested.

Line 67: can you add from this paper also- Citation Kirstahler P, Teudt F, Otani S, Aarestrup FM, Pamp SJ. 2021. A peek into the plasmidome of global sewage. mSystems 6: e00283-21. https://doi.org/10.1128/mSystems .00283-21.

Response:

The reference was cited in the description.

Line 160- is it the most predominant subclonal type?

Response:

Besides the most predominant ST131, other ST isolates were found in this study.

Table 1- what are these a,  b citations please can you open them? Do not use abbreviations in the table open EPO and CPO in the footnote.

Response:

a [25,26] indicated the previous reports about KPC-2-prooducing bacteria,

b [27] indicated the previous reports about NDM-5-producing bacteria.

We have reported the KPC-2 or NDM-5-positive WWTP-effluent isolates i.e. the complete genome sequences of KPC-2-positive Klebsiella pneumoniae GSU10-3 [25], KPC-2-positive Aeromonas hydrophila GSH8-2 [26], KPC-2-positive Aeromonas caviae GSH8M-1 [26], and NDM-5- and CTX-M-55-coproducing E. coli GSH8M-2 [27] were determined.

Figures 1-4: seems like a little crowded, if any of your figures are not relevant in the main text you can push them in supplemental (it is just optional).  If some of them are based on secondary reads from databases.

Response:

We would like to present all figures in the manuscript, if there is enough space on the paper within the word limits.

He concluded WWTP effluent is expected to identify the presence of undetected nosocomial infection, ongoing in the community. But, if we want to see from the perspective of wastewater-based surveillance, it supposes to use wastewater influent. Please mention that somewhere in the discussion or the introduction (here this study can be a potential reference DOI: 10.3389/fmicb.2022.887888.

Response:

At first, we used WWTP effluents, not but its influents.

Most people are rarely exposed by the WWTP influent, whereas urban WWTP effluents, even with proper treatment, may cause AMR burden with a high frequency of acquired ARGs in the environment such as river, lake and ocean. Thus, we considered that the effluent is one of the best environmental sources to investigate the actual risk assessment for people nearby WWTP.